# Automating the quantification of coastal change using historical aerial photography: A case study along the coastline of county Cork, Ireland

Emma Chalençon[1,2] ⬤, Fiona Cawkwell[1], Michael O'Shea[2] and Jimmy Murphy[2]

[1]Department of Geography, University College Cork, T12K8AF Cork, Ireland and [2]MaREI Centre, Environmental Research Institute (ERI), University College Cork, P43 C573 Cork, Ireland

## Research Article

aerial photography; coastal monitoring; shoreline change; vegetation line; colour vegetation indices

**Corresponding author:**
Emma Chalençon;
Email: emma.chalencon@ucc.ie

## Abstract

Coastlines worldwide are coming under increasing pressure due to climate change and human activity. Data on shoreline change are essential for coastal managers and when no long-term monitoring programs are implemented and shoreline change is typically on the order of less than 1 m/yr., as observed in Ireland, aerial photography is the most valuable source of information. A well-established literature exists for automated vegetation extraction from digital images based on the near infrared reflectance, but there is less research available on spectrally limited colour photography. This study develops a methodology for automating vegetation line extraction from a series of historical aerial photography of the Cork coastline in the South-West of Ireland. The approach relies on the Normalised Green–Blue Difference Index (NGBDI), which is versatile enough to discriminate disparate coastal vegetation environments, at different resolutions and in various lighting and seasonal conditions. An iterative optimal threshold process and the use of LiDAR ancillary datasets resulted in an automated vegetation line measurement with uncertainties estimated to be between 0.6 and 1.2 m. Change rates derived from the vegetation lines extracted present uncertainties in the range of ±0.27 m/yr. This robust and repeatable method provides a valuable alternative to time-consuming and subjective manual digitisation.

## Impact statement

Coastlines worldwide require effective management, and accurate, timely data on shoreline movements are an indispensable prerequisite to inform the decisions made by coastal managers. Field coastal monitoring requires considerable human resources, it is spatially limited and time-consuming, but significantly it cannot be done retrospectively. In places where no such programmes have been undertaken, Earth Observation satellite data can be invaluable in capturing temporal changes. But where shoreline changes, or movement of the vegetation line, is typically on the order of less than 1 m per year, as observed in Ireland, aerial photography is the most valuable source of regional to national scale information. While it is common practice, manual digitisation of shorelines is subjective and time consuming. Substantial literature is available on automated vegetation feature extraction using near-infrared reflectance but, research on more spectrally limited RGB (red-green-blue) colour photography, commonly acquired by aerial platforms, is limited to very high-resolution Uncrewed Aerial Vehicles (UAV) photography. In this paper, we demonstrate the viability of automated shoreline detection on aerial orthophotography making use of Colour Vegetation Indices developed for UAV photography. Historical archives of aerial photography are unevenly stocked with photography of varying quality and acquisition conditions, alongside limited spectral content, making them challenging datasets to handle, but the methodology developed has proved versatile enough to perform well at different resolutions, and in different lighting and seasonal conditions, effectively discriminating diverse coastal vegetation environments. This research provides a robust and repeatable method to extract shoreline change information from data-limited archives.

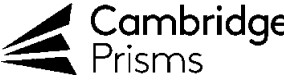

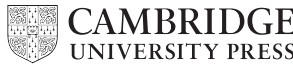

## Introduction

Following a worldwide pattern (UNEP, 2017), the highest concentrations of population and activity in the Republic of Ireland are found in coastal areas with 1.9 million people residing within 5 km of the coast, representing 40% of its population (CSO, n.d.). Human activities coupled with a changing climate, associated with rising sea levels and an increase in storminess, impact shoreline movements and can have major detrimental effects. Coastal erosion and flooding can eventually lead to a loss of habitats and ecosystems, damage to a range of infrastructure, and disruption to social and economic systems (IPCC, 2018). Coastlines

worldwide require ongoing effective management, and accurate, timely data on shoreline movement are an indispensable prerequisite to inform the decisions made by coastal managers.

In recent years in Ireland there has been a growing interest from stakeholders for accurate data on coastal change to better address challenges faced by populations, infrastructures, and ecosystems. This need was underscored in the Report of the Inter-Departmental Group on National Coastal Change Management Strategy, published in October 2023, which identified deficiencies, including the lack of monitoring along the majority of the national coastline (Department of Housing, Local government and Heritage and the Office of Public Works, 2023). The most recent national coastal erosion assessment undertaken in Ireland was the Irish Coastal Protection Strategy Study (RPS/ICPSS, 2011). The shoreline position was retrieved from manual digitisation of aerial photography at different dates between 1973 and 2006 (RPS/ICPSS, 2011). Annual retreat rates were derived assuming a linear retreat process, and the change in position of the shoreline was measured at a very coarse resolution of 1 km. This analysis is now outdated and must be extended in time to account for shoreline change which has happened since 2006. Despite these limitations and the dataset's focus on identification of retreating coastal segments, it has been the only quantitative reference used by local authorities in Ireland since 2011 (Flood and Schechtman, 2014; McKibbin, 2016; Lawlor and Cooper, 2024).

Consistent archives of coastal movements over multiple decades are rare. In the United Kingdom, the East Riding Regional Coastal Monitoring Programme established in the late 1990s, with collections of beach cross-profiles at 75 different points along the coast every six months, is an example of best practice (East Riding of Yorkshire Council, 2006). Moreover, annual aerial photographs from the past two decades, available through the Channel Coast Observatory (CCO), provide a valuable resource for large-scale shoreline change analysis, complementing localised and resource-intensive field monitoring efforts. In places where no monitoring programmes have been undertaken, maps are invaluable for shoreline change analysis due to their historical significance. However, historical maps in Ireland are infrequent and often lack precision, preventing their inclusion in the study and necessitating a reliance on aerial photography. Ireland holds an archive of national photography captured periodically since 1995. The spatial and spectral resolution of aerial photography acquired worldwide is very varied, but typically, the older the aerial photography, the less detail is available, with the first national campaign in Ireland only acquiring panchromatic photography for example. Aerial photography acquired in three or more spectral bands are now more common, and in Ireland, national photography was acquired in the Red, Green, and Blue (RGB) parts of the electromagnetic spectrum up until 2013, after which the Near Infrared (NIR) was included.

In this study, shoreline will refer to the dynamic boundary where the land meets the sea, a line subject to change from natural and human influences. The coastline encompasses the entire length of land along the sea. Shoreline detection techniques are generally classified into datum-based methods, which utilise LiDAR or other elevation capture technologies to create digital terrain models (DTMs), and proxy-based methods (Pollard, Brooks, and Spencer 2019). Datum-based methods are limited by infrequent image capture and inconsistent spatial coverage (Pardo-Pascual et al. 2018), limitations which apply to Cork. Proxy-based methods rely on the detection of visible indicators whether they are geomorphological, vegetation, water or human features (Toure et al. 2019). The most frequently identified shoreline indicator from optical images is the instantaneous waterline, as it is the most visually discernible feature (McAllister et al. 2022). However, to use instantaneous waterlines as indicators of shoreline change, they must be corrected using estimates of beach slope and tidal height timeseries, which can be challenging to obtain in areas with observation gaps (Muir et al., 2024), such as along the Cork coastline. On the contrary, the seaward edge of stable coastal vegetation, the vegetation line, serves as a less variable shoreline proxy (Pollard et al., 2020), effectively capturing changes without the bias introduced by tidal stages (Toure et al., 2019). While the vegetation line may vary seasonally, it was selected as shoreline proxy given the study area data limitations. Although this proxy is ineffective on artificial or hard cliff coasts, it is a valuable indicator of shoreline change in soft, sandy environments, such as Cork, where storm energy gradients drive coastal dynamics (Pollard et al., 2020; Devoy, 2008). Additionally, remote sensing techniques for mapping vegetation have a well-established research history (Ustin and Gamon, 2010). Vegetation is traditionally mapped with indices using NIR and red reflectance. The normalised difference vegetation index (NDVI) is the most widely used metric when it comes to quantifying the health and density of vegetation (Huang et al., 2021). However, historical aerial photography do not commonly include NIR information.

The use of Colour Vegetation Indices (CVI) based only on RGB data grew with the popularisation of UAV research. Most CVIs were thereby designed for centimetre scale resolution photography. UAVs can play a significant role in monitoring and managing coastal ecosystems (Joyce et al., 2023), however they cannot be acquired retrospectively to calculate historical change rates. This research proposes a methodology to adapt the use of UAV-CVIs to much coarser historical aerial photography for the purpose of historical vegetation line identification.

## Study area and data

### Study area

The coastline of Ireland is very irregular with a bay-headland configuration resulting from a high wave energy regime. Cork in the South-West of the Republic of Ireland has 1,094 km of coastline (Figure 1), and it is the county recording the highest proportion of its population living within 100 m of the coast (CSO, n.d.). Cork has 422 km of soft sandy coastline, and 91 km are at risk of erosion based on the results of the Ecopro (1996) and Eurosion (Salman et al., 2004) projects. The eastern part of Cork's coastline is highlighted as more vulnerable due to its geomorphological attributes and the higher recorded erosion rates in that area.

The methodology proposed to extract vegetation lines from historical aerial photography and quantify shoreline change is applied to the entire Cork coastline. However, five sites along the coast have been chosen to validate the results of this study (Figure 1). From East to West, Pilmore and Garryvoe beaches were selected as two of the sites recording the highest retreat rates in County Cork. Inchydoney and Owenahincha are two West Cork beaches with large dune systems which make them very popular beaches. Finally, Garinish Bay hosts three small sandy coves on the Beara Peninsula in the western part of County Cork.

### Aerial photography

Tailte Éireann is the Irish agency in charge of national mapping. They completed their first full coverage of the Republic of Ireland

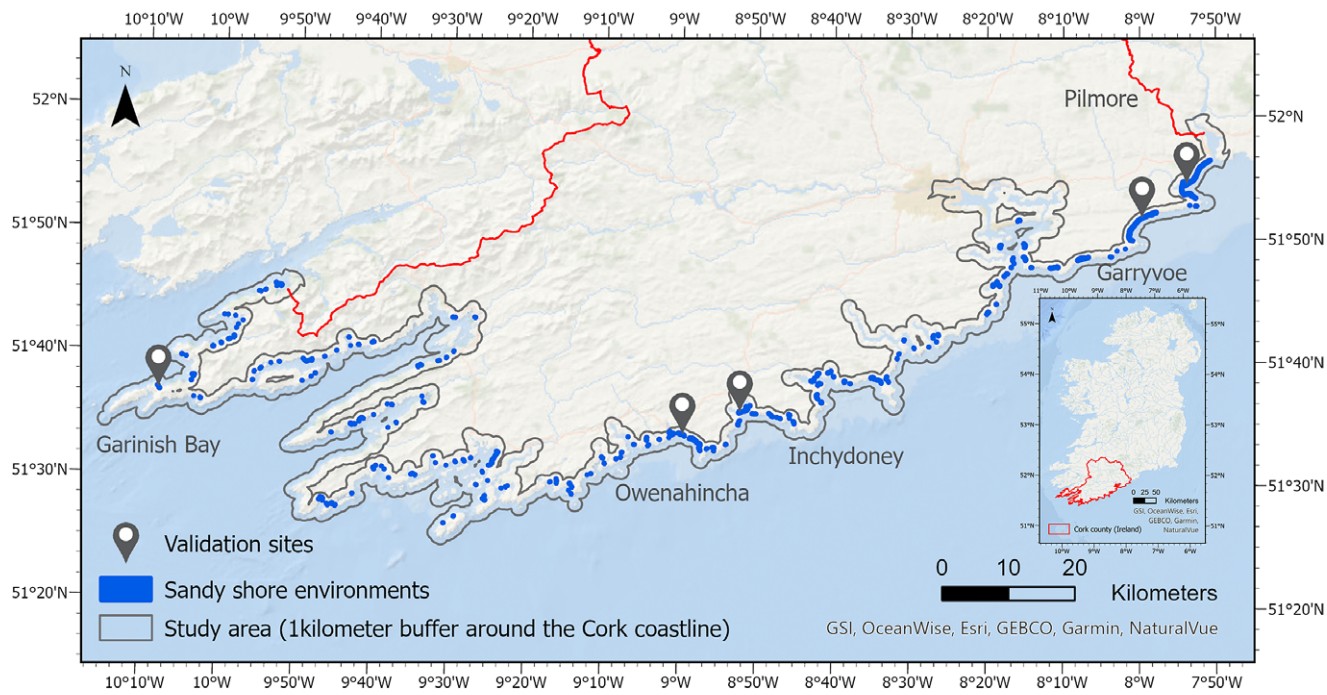

**Figure 1.** Study area's location in Ireland and validation sites along Cork coastline with sandy shore environments highlighted in blue. Inset shows the location of County Cork within the island of Ireland.

**Table 1.** Photography used in the analysis

| Dataset name | Date for Cork | Spatial resolution (m) | Spectral resolution | Positional accuracy uncertainty (m) | Optimal threshold |
|---|---|---|---|---|---|
| MapGenie Photography (1996–2000) | 07/2000 | 1 | RGB | < 0.5 | 0.1 |
| MapGenie Photography (2004–2006) | 07/2005 | 1 | RGB | < 0.5 | 0.1 |
| MapGenie Photography (2005–2012) | 11/2011 03/2012 | 0.25 | RGB | < 0.5 | 0.08 |
| MapGenie Photography (2013–2018) | 04/2015 06/2018 | 0.25 | RGB + NIR | < 0.5 | 0.15 0.06 |
| Coastal Aerial LiDAR Survey 2021[1] | Autumn 2021 | 0.10 | RGB + NIR | < 0.1 | 0.15 |

[1]This dataset has been produced by the Office of Public Works and is only covering the South-Western coast of Ireland. It only covers the western part of the study area, west of Cork harbour.

RGB aerial photography dataset in 2000. From 2000 onwards, national coverage orthophotography datasets have been delivered periodically with increasing spatial and spectral resolutions (Table 1).

Since field monitoring data exist only for a few sites for a single season and satellite imagery are unsuitable due to the magnitude of change observed, aerial images are the most valuable—and invariably the only—source of historical coastal positions in Ireland. Nevertheless, working with aerial photography in Ireland can be highly challenging. Aligning the availability of survey aircraft on the island with cloud-free weather conditions at times of high sun angles in the summer season for the whole country is nearly impossible. Achieving national coverage may entail flights spanning up to 5 years apart, occurring from March to November. The exact time and date of acquisition for each photography is not always available as these datasets have been produced by different contractors over the years with different procedures and metadata requirements.

Aerial photography are orthorectified by the data provider, with each pixel having x and y co-ordinates representing its position on the ground so that accurate measurements can be taken from them, but the uncertainty varies between the datasets (Table 1, Column 5: "Positional accuracy uncertainty (m)").

### Complementary datasets

Seaweed washed ashore and low-tide shallow waters might have similar spectral signatures in the visible wavelengths to growing vegetation, therefore ancillary datasets have been used to refine the study area and mask areas prone to misclassifications in low-lying areas. LiDAR coverage of the Cork coastline is limited in frequency and spatial coverage, but several datasets are available, each covering different sections of the coastline; the eastern Cork coastline was surveyed as part of the Office of Public Works (OPW) Blom Coastal Survey in 2006–2007, Cork Harbour, as part of the OPW Flimap Survey in 2007 and the OPW Coastal Aerial LiDAR survey covered the western part of the county's coastline in 2021. To mask out low-lying areas where misclassification issues can arise, all areas under 2 m of elevation to the Malin Head datum on the different LiDAR Digital Surface Models

(DSMs) were merged to create a low-lying areas mask. This threshold was determined through an iterative optimal thresholding process, aimed at masking as much low-lying area as possible without compromising the accommodation space for the vegetation line.

The choice of the vegetation line proxy for shoreline position is only relevant for soft coasts, which are more vulnerable to change over time, and is not suitable for hard or artificial coasts unless they are vegetated seaward. The previous coast classification work achieved by the Eurosion (Salman et al, 2004) and the ICPSS (RPS, 2011) projects served as guidelines to identify soft coastal segments. These were further refined using the National Land Cover Map (NLCM), created by Tailte Éireann and the Environmental Protection Agency (EPA), and visual inspection using the study photography database. This work resulted in a sandy shore environments zone.

## Methods

### Selecting a suitable CVI

The use of vegetation indices is a common practice in remote-sensing studies, as they minimise the influence of distorting factors (Ruiz, 1995) as well as combining and maximising information from specific bands or parts of the electromagnetic spectrum. Several CVIs based on colour RGB photography have been proposed to identify vegetation, primarily for data from UAVs carrying RGB cameras. These CVIs include the normalised green-red difference index (NGRDI) (Torres-Sanchez et al., 2013), the visible-band difference vegetation index (VDVI) (Wang and Myint 2015), the normalised green-blue difference index (NGBDI) (Wang and Myint 2015) and the Red-green-blue vegetation index (RGBVI) (Bendig et al. 2015).

The index chosen had to be versatile enough to perform well at different resolutions, and in different lighting and seasonal conditions to discriminate very different vegetation environments. The three coves of Garinish Bay backed by grass vegetation and the dunes from the sandspit of Inchydoney (Figure 1) were chosen to test five different indices. The binary classifications of vegetation or no vegetation resulting from the different indices were assessed using the widely recognised overall accuracy metric, calculated as the total number of correctly classified pixels divided by the total number of pixels in the reference data. Using the 2000 photography (1 m spatial resolution), the NGBDI (Equation (1)) outperformed the NGRDI by 30%, the RGBVI by 5% and the VDVI by 9%, achieving 89% classification accuracy when compared with manual photointerpretation. Using the 2018 photography (0.25 m spatial resolution), the NGBDI was once again the best performing index with an accuracy of 96%, similar to the 95% performance of the RGBVI. Since the 2018 photography also contained NIR data, the performance of the NGBDI was compared to that of the commonly used Normalised Difference Vegetation Index (NDVI), with a very similar accuracy of 94% achieved. Using the 2021 photography (0.1 m spatial resolution), all indices performed similarly with accuracies of 97–98%, with the exception of the NGRDI, which had an accuracy of 79%. After testing the different indices, the one which performed most consistently across the different photography sets, gave the best statistical accuracy and generated the most coherent vegetation line was the NGBDI (Eq. 1).

$$NGBDI = (Green\text{-}Blue)/(Green + Blue) \qquad (1)$$

The study's regional scope, the limited uniformity of sandy environments along the Cork coastline, and large variations in data acquisition conditions precluded use of image classification methods. The extensive training required, which would have to be undertaken for each image set, would have negated the time-saving benefits of developing an automated approach. To objectively differentiate between vegetation and non-vegetation pixels for the varied environmental and acquisition conditions, an iterative optimal threshold process was implemented, with different NGBDI thresholds tested, by visual examination of the spectral signature of nearby pixels and defined according to the resolution of the dataset as well as the seasonality of the acquisition date.

At 1 m resolution, each pixel tends to represent a homogeneous area. With clear boundaries and fewer mixed pixels, the distinction between features is more pronounced and higher thresholds can be applied. A threshold value of 0.1 was chosen for both the 2000 and 2005 datasets.

At 0.25 m resolution, more details are captured in the photography. Nevertheless, the increased level of detail may not fully distinguish boundaries with intricate details and with more mixed pixels, it becomes challenging to precisely delineate boundaries. As a result, a more permissive threshold was needed to ensure that features of interest were captured accurately. Therefore, thresholds of 0.08 and 0.06 were chosen for the 2011–2012 and the 2018 datasets respectively. The 2015 photography were treated separately from the 2018 photography given the season difference (April 2015 versus June 2018). The 2015 photography covers the Eastern part of Cork coastline, which is more homogeneous with linear beaches backed by grass vegetation and no large dune systems. In April, grass is reaching its growing peak, and its green reflectance is very distinctive. These conditions justify the choice of a higher 0.15 threshold for the 2015 photography.

At 0.1 m resolution, boundaries are more clearly defined, and features can be easily captured on the 2021 photography. As a result, a higher threshold of 0.15 was applied to this dataset.

### From a binary photograph to a vegetation line

Applying the selected threshold to the NGBDI output resulted in binary outputs of vegetation pixels and background pixels, which had to be converted into a line feature for subsequent input to the Digital Shoreline Analysis System (DSAS) (Himmelstoss et al., 2021). The binary images were first polygonised then simplified using a double buffer process. First, a positive buffer is applied, extending the vegetation polygon by a distance corresponding to the photography's resolution. As a second step, a negative buffer is performed, contracting the vegetation area by the same distance. This process helps smooth the vegetation edge, simplifying its geometry.

Polygons under 8m$^2$ were usually identified as seaweed residuals or small patches of vegetation not suitable to be integrated into the vegetation line. Based on this observation, all polygons under 8m$^2$ and whose centroid lay within the National Land Cover Map's 'Exposed Sediments' class were deleted. The remaining polygons were agglomerated using an agglomeration distance of 10 m, a minimum area of 80m$^2$ and a minimum hole area of 10,000m$^2$. They were finally converted into line features, and only lines within 50 m of the initial 2000 vegetation line were kept for the DSAS analysis. Vegetation lines were thus created along the Cork

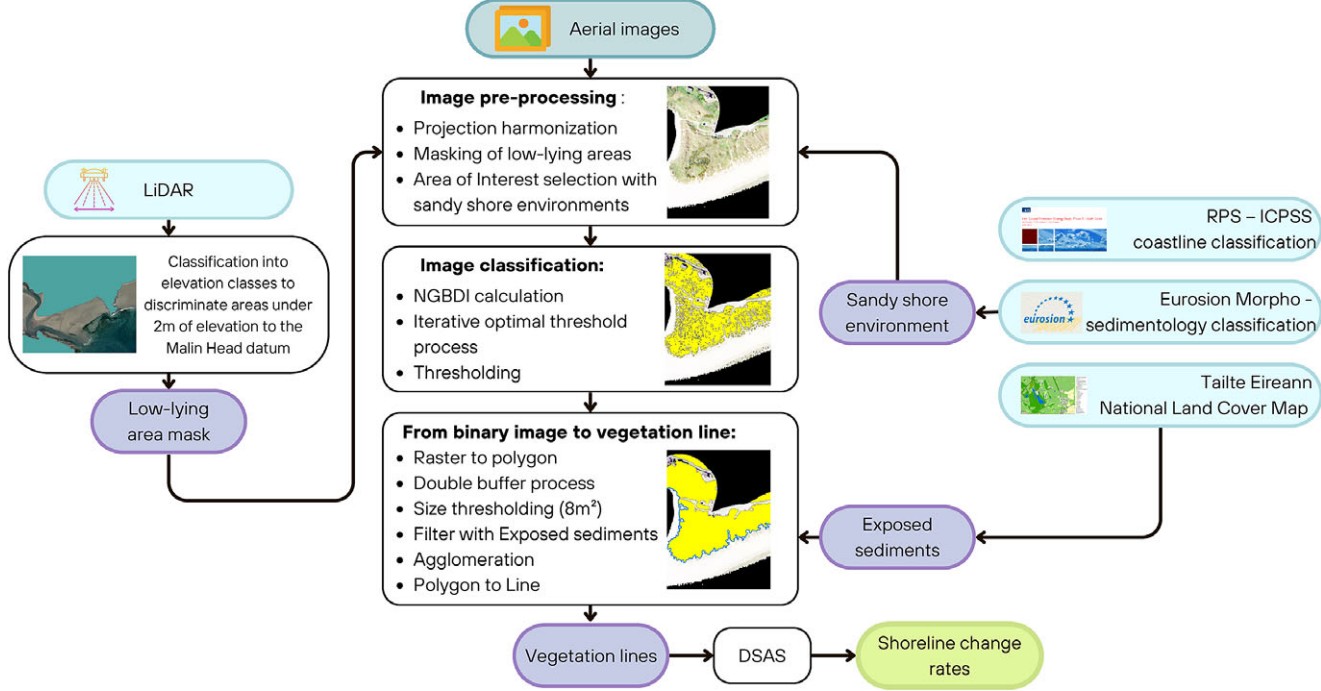

**Figure 2.** Workflow schematic from historical aerial photography to shoreline change rates.

coastline as proxies of shoreline position in 2000, 2005, 2011 or 2012, 2015 or 2018, and 2021. The full workflow can be seen in Figure 2.

### DSAS analysis

The DSAS is a freely available software application that works within the Esri ArcGIS software and calculates change statistics for a time series of shoreline vector spatial features (Himmelstoss et al., 2021). The DSAS first requires a baseline to build transects along which rates of change will be calculated. For consistency of measuring change using the data available to this project, the 2000 vegetation line was selected. This baseline was categorised as midshore, enabling transects to account for both retreat and accretion. The maximum search distance was set to 30 m to allow for large movements observed at sand spits, but without transects intersecting each other in smaller coves. Transects were located at 10 m intervals and no smoothing distance was applied, as it tended to place transects inappropriately parallel to the baseline. No manual editing or omission of transects crossing the shorelines at oblique angles was performed, in order to make the process as automated as possible and avoid manual intervention. This approach was feasible because, unlike the overall sinuous Cork coastline, the soft shore segments are relatively straight. All statistics available were calculated for each transect. The Shoreline Change Envelope (SCE) represents the distance between the most seaward and the most landward shorelines that intersect a specific transect. The end point rate (EPR) is calculated by dividing the SCE by the time elapsed between the first and last dated shorelines that intersect a given transect. A linear regression rate-of-change (LRR) statistic is calculated by fitting a least-squares regression line to all shoreline points for a transect (Himmelstoss et al., 2021).

### Validation

As no pre-existing dataset was available to validate the vegetation lines it was decided to manually digitise vegetation lines for each available year at the five validation sites (Figure 1). Points were generated every 25 cm along the manually digitised vegetation lines, and at each point the distance between the manually and automatically derived lines was recorded to calculate the Mean Absolute Error (MAE).

## Results

### Validating the automated detection of vegetation lines

Vegetation lines were generated at every soft-shore site along the Cork coastline for 2000, 2005, 2011 or 2012, 2015 or 2018, and 2021 (Figure 3). The OPW Coastal Aerial Survey acquired in 2021 is only available for sites West from Cork Harbour, therefore, five vegetation lines were produced for the three sites West of Cork Harbour (Figure 3) and only four lines for the two sites East of Cork Harbour (Figure 1). The Mean Absolute Error (MAE) and its respective standard deviation for each site is recorded in Table 2.

The July 2000 vegetation lines record MAEs below one pixel across all sites, except for Inchydoney, where the MAE slightly surpasses 1 m at 1.09 m due to some embryo dunes with vegetation patches being omitted (Figure 4 - A). Given the relatively coarse resolution of the orthophotography, the results accurately capture the vegetation lines at each site.

The results for the July 2005 vegetation lines are similar, with MAEs below one pixel across all sites. The best outcomes are observed at Garryvoe beach with a MAE of 0.57 m coupled with a minimal standard deviation of 0.64 m (Figure 4 – B). Garryvoe beach is backed by glacial tills covered by agricultural fields. In July, these grasslands display a very distinctive green reflectance,

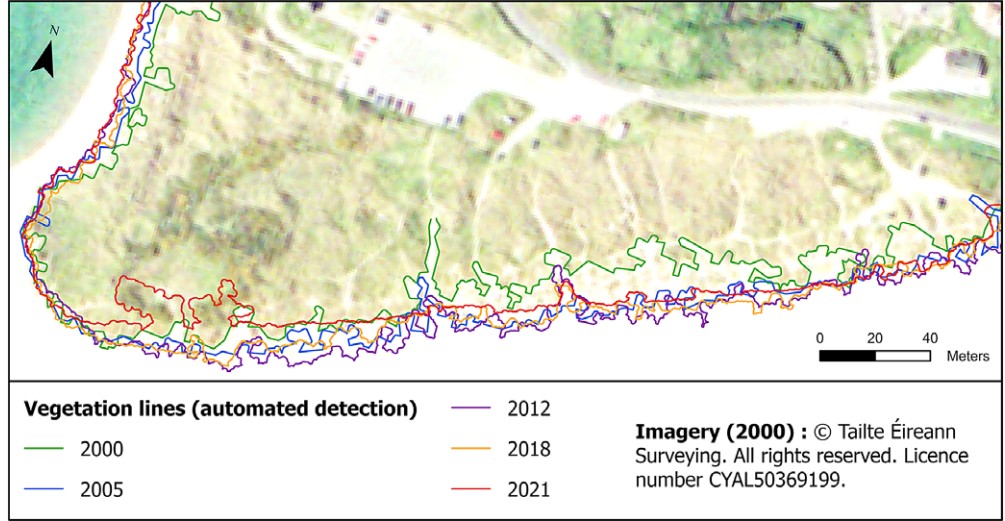

**Figure 3.** Vegetation lines produced with the automated method at Inchydoney beach between 2000 and 2021.

making it relatively easy to distinguish them from the sandy beach.

At the 0.25 m spatial resolution of the November 2011 and March 2012 photography, several sites show their largest MAEs. When remote sensing data are captured at a higher resolution, it means that smaller and more complex details of the landcover are captured. However, there is a critical point where the resolution might not be sufficient to capture the full complexity of the landcover features. Real-world features are indeed often characterised by fractal patterns that exhibit details at various scales. A discrepancy between the resolution and the complexity of the landcover features may lead to misinterpretations or incomplete delineation of landcovers. Inchydoney and Garryvoe beaches have MAEs slightly over 1 m, and Owenahincha beach records a 1.66 m MAE with a large standard deviation at 2.16 m. For all these sites, the photography have been acquired in March, which is quite early in the spring season, and the vegetation is not yet at its greenest, adding complexity to its detection.

At Owenahincha Beach in 2012, the vegetation line alternates between the most seaward vegetation and more landward vegetation similar to that observed at Inchydoney beach. The algorithm misses the pioneer marram grass, which has low contrast with the sand. This occurs at a resolution that introduces additional inaccuracies, complicating precise boundary delineation. It is important to note that although using manual digitisation as a validation source in remote sensing is a legitimate approach, especially when alternative validation sources are unavailable, it is subjective and may introduce its own set of inaccuracies.

The results obtained for the national orthophotography mosaic 2013–2018 are quite heterogeneous. Just like Garryvoe beach, Pilmore is a long linear beach backed by grasslands and 2015 is the year where its MAE is the lowest at 0.38 m, or under two pixels of this dataset (Figure 4 - D). Nevertheless, the issue related to embryo dunes and pioneer vegetation patches is still present at Inchydoney beach in 2018, giving a MAE close to 3 m (Table 2).

The last set of photography for 2021 is only available for the three sites West from Cork Harbour. The spatial resolution is enhanced to 0.1 m and the overall results are the best across the different years. MAEs are below 0.75 m across all sites, and below 0.6 m at the three coves of Garinish Bay (Figure 4 - E). The

improved resolution captures additional complexity and intricate details, allowing better differentiation between features, and reaching the fractal analysis critical point where the complexity can fully be captured.

### *Validating the resulting change rates*

Although the validation of the extracted vegetation lines' position for each year is critical, it is crucial to establish the degree to which positional errors, specific to each year, impact the resultant change rates. For each of the five validation sites, a DSAS analysis was performed using the manually digitised vegetation lines and compared with the DSAS analysis based on the lines extracted using the automated method (Figure 5).

The average MAEs for End Point Rates across all sites is 0.24 m/yr. (Table 2). Given this result, EPRs within the range ± 0.25 m/yr. may indicate a tendency towards stability rather than change. When shoreline change lies within the error bounds, it is not possible to indicate directional shoreline change (Pollard et al., 2020).

The dune system at Owenahincha beach shows MAEs around 0.25 m (Table 2, Figure 5 – C). The difference between the average rates calculated using both methods at Owenahincha is under 0.05 m/yr. Although it was one of the sites that showed the largest errors when considering the positional accuracy of the individual automated vegetation lines, the embryo dunes omitted one year are either fully integrated into the dune system or washed away on the next photography, making little difference to the overall rates of vegetation line change.

Pilmore and Garinish Bay record the lowest MAEs (Table 2, Figure 5 – D & E), and average rates at these sites show good agreement between the automated and manual approaches, with differences of less than −0.05 m/yr. for Pilmore and 0.09 m/yr. for the three coves of Garinish bay (Table 2). At Garryvoe beach, MAEs reach 0.37 m (Table 2) and even though retreat is indicated by both approaches, the difference in the average rates is 0.27 m/yr. (Table 2, Figure 5 – B). Unlike other sites, Garryvoe beach is backed by agricultural land. In some seasons some of these fields were not vegetated and no vegetation line could be extracted for the most western field on the 2011 and 2015 photography covering Garryvoe

**Table 2.** Accuracy of the vegetation lines produced with the automated method (m) as well as the accuracy of the change rates (m/yr) derived from these lines

| Sites | Years | Mean Absolute Error (m) | Standard Deviation (m) | EPR Mean Absolute Error (m/yr) | Average EPR (manual lines, m/yr) | Average EPR (automated method, m/yr) | Average EPR difference (m/yr) |
|---|---|---|---|---|---|---|---|
| Inchydoney | 2000 | 1.09 | 1.72 | | | | |
| | 2005 | 0.78 | 0.91 | | | | |
| | 2011–2012 | 1.06 | 1.16 | **0.18** | 0.14 | 0.11 | −0.03 |
| | 2015–2018 | 2.92 | 2.84 | | | | |
| | 2021 | 0.74 | 1.13 | | | | |
| | Average | **1.42** | **1.99** | | | | |
| Garryvoe | 2000 | 0.76 | 0.83 | | | | |
| | 2005 | 0.57 | 0.64 | | | | |
| | 2011–2012 | 1.05 | 1.48 | **0.37** | −0.62 | −0.35 | 0.27 |
| | 2015–2018 | 0.68 | 1.44 | | | | |
| | 2021 | | | | | | |
| | Average | **0.74** | **1.11** | | | | |
| Owenahincha | 2000 | 0.72 | 0.77 | | | | |
| | 2005 | 0.90 | 1.20 | | | | |
| | 2011–2012 | 1.66 | 2.16 | **0.22** | 0.61 | 0.59 | −0.02 |
| | 2015–2018 | 1.10 | 1.64 | | | | |
| | 2021 | 0.44 | 0.82 | | | | |
| | Average | **0.95** | **1.46** | | | | |
| Pilmore | 2000 | 0.98 | 1.5 | | | | |
| | 2005 | 0.89 | 1.28 | | | | |
| | 2011–2012 | 0.71 | 1.06 | **0.15** | −0.34 | −0.40 | 0.05 |
| | 2015–2018 | 0.38 | 0.74 | | | | |
| | 2021 | | | | | | |
| | Average | **0.71** | **1.16** | | | | |
| Garinish bay | 2000 | 0.96 | 0.83 | | | | |
| | 2005 | 0.88 | 0.79 | | | | |
| | 2011–2012 | 0.90 | 1.25 | **0.09** | −0.18 | −0.10 | 0.09 |
| | 2015–2018 | 1.04 | 1.64 | | | | |
| | 2021 | 0.60 | 1.27 | | | | |
| | Average | **0.87** | **1.24** | | | | |
| Average across all sites | 2000 | 0.88 | 1.19 | | | | |
| | 2005 | 0.76 | 0.97 | | | | |
| | 2011–2012 | 1.07 | 1.48 | **0.24** | −0.10 | −0.03 | 0.08 |
| | 2015–2018 | 1.24 | 1.83 | | | | |
| | 2021 | 0.59 | 1.07 | | | | |
| | Average | **0.99** | **1.53** | | | | |

beach, which explains why some lines erroneously veer north at the west end of Figure 5 – B. As the final photography for this analysis is from 2015, a large error for this date can have greater consequences for the final EPR of this specific part of the vegetation line. MAEs for the rest of the vegetation line at Garryvoe beach show good agreements with the change rates derived from manual digitisation (Figure 5 – B).

## Discussion

### *A robust alternative to manual digitisation*

Historical aerial photographs are often the only available evidence of past coastal positions, but their disparate quality, conditions of acquisitions, positional accuracy, and limited spectral content make them challenging datasets to work with. This explains why many

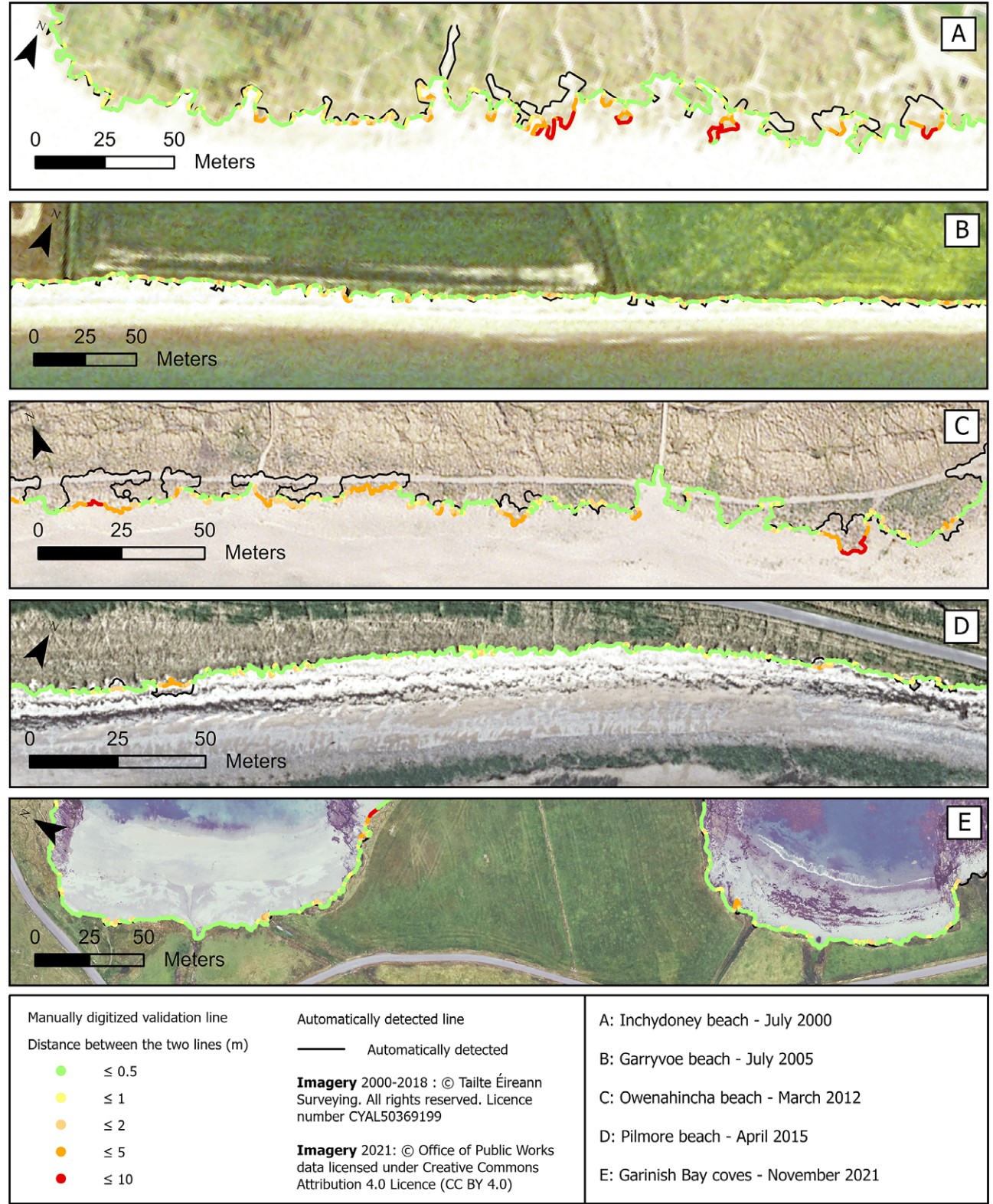

**Figure 4.** Validation of vegetation lines produced with the automated method, using custom manually digitised lines at Inchydoney beach in July 2000 (A), at Garryvoe beach in July 2005 (B), at Owenahincha beach in March 2012 (C), at Pilmore beach in April 2015 (D) and at Garinish bay coves in November 2021 (E). At Owenahincha Beach in 2012, the vegetation line alternates between the most seaward vegetation and more landward vegetation because the algorithm misses the pioneer marram grass, which has low contrast with the sand. This occurs at a resolution that introduces additional inaccuracies, complicating precise boundary delineation.

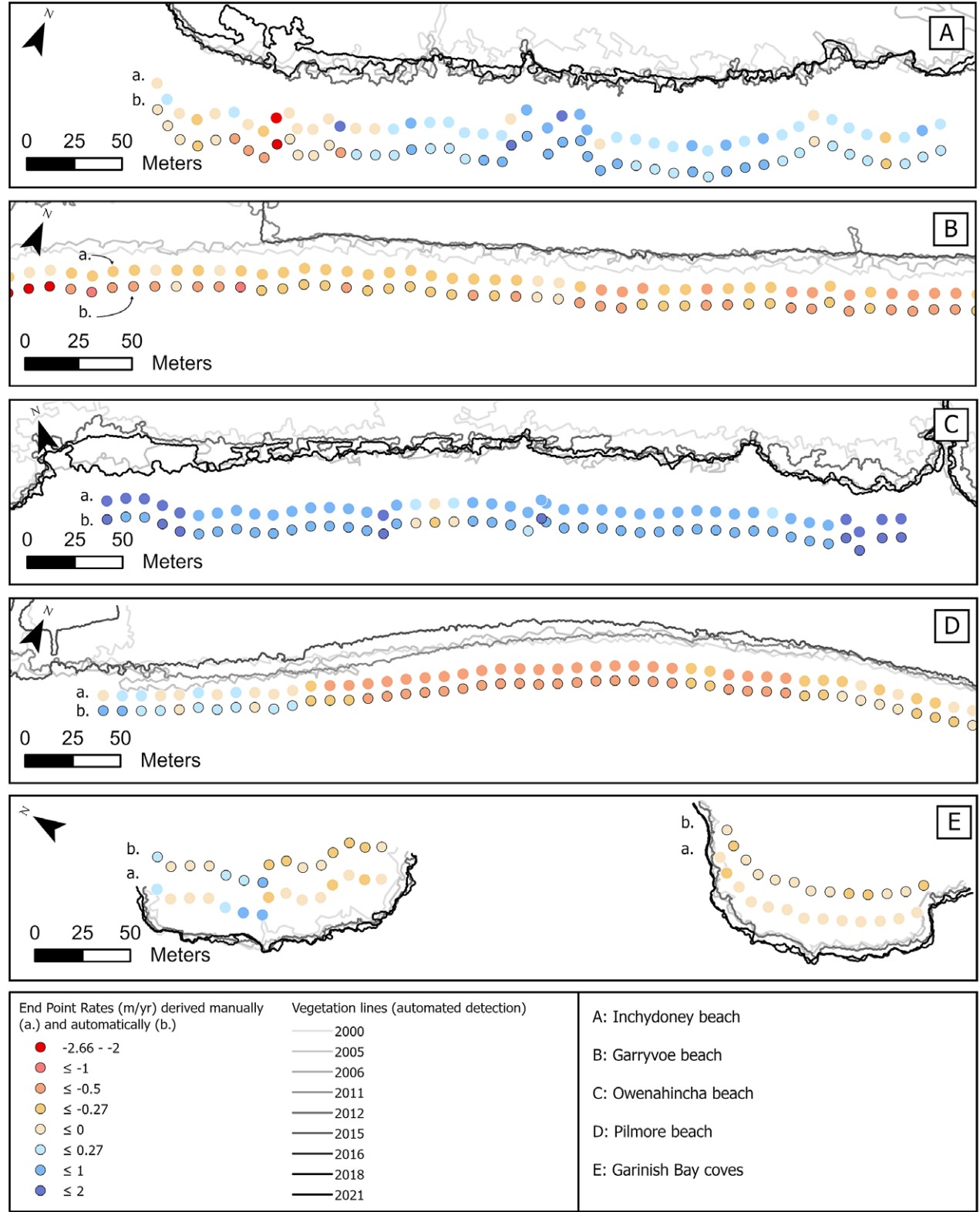

**Figure 5.** Validation of End Point Rates (EPR) derived automatically (b) against manually derived rates (a) at Inchydoney Beach (A), Garryvoe Beach (B), Owenahincha Beach (C), Pilmore Beach (D), and Garinish Bay Coves (E). The evolution of the vegetation line is illustrated with periodic lines automatically produced from the available aerial image datasets.

studies have relied on manual digitisation. The last three national or regional studies on coastal change in the Irish context made this choice; the OPW (RPS/ICPSS, 2011), the Geological Survey Ireland (GSI) (GSI, 2023) and the Northern Ireland Historical Shorelines

Analysis (NIHSA) project (Grottoli et al., 2023). In a publication from 2021, Fabbri et al. report maximum digitising errors arising from subjectivity of 0.3 m for the Dune Foot Line and 0.85 m for the Stable Vegetation Line on UAV photography with a spatial

resolution of 2–4 cm. The GSI's National Assessment of Shoreline Change Report published in 2023, reports uncertainties in vegetation line measurements of 1 m, for the 2000 and 2005 datasets and 0.5 m, for the 2005–2012 and 2013–2018 datasets. Although the reported uncertainty for the two latter datasets (0.5 m) is slightly better than the 0.99 m MAE given for the method presented here, the uncertainty for the first two is comparable. Notably, the results from the GSI correspond to the digitisation of County Dublin's coast where beaches tend to be longer and straighter than the indented and varied coastline of County Cork. It is important to emphasise the subjective nature of manual digitisation, whether employed for a final product or validation purposes, especially in environments involving fragmented vegetation lines in dune systems. The accuracy of the position or even the existence of a true vegetation line may be subject to diverse interpretations from experts of equal knowledge.

Prior to this work, Cork County Council relied exclusively on ICPSS outcomes to guide discussions and management of coastal risks. Of the five validation sites, only two had available outputs. For Garryvoe, the ICPSS divided the area into two segments: the western two-thirds indicated an erosion rate of 0.33 m/yr., while the eastern third showed no erosion (0 m/yr). In contrast, the automated method used in this study returned an average EPR of −0.85 m/yr. for the western segment and − 0.25 m/yr. for the eastern end. Pilmore Beach was covered by a single ICPSS segment, indicating an erosion rate of 0 m/yr., whereas the automated method revealed an average EPR of −0.40 m/yr.

Regarding sites not flagged by the ICPSS, no clear dynamic patterns were observed at Garinish Bay coves, as the rates fall within the margin of error. Owenahincha serves as an example of best practice. After experiencing severe erosion in the 1970s (Mullane and MacSweeney 1977), the introduction of gabions, dune reshaping, and replanting stabilised the area, and this study reveals the steadily advancing vegetation line, confirming the resilience of the managed dunes. While local concerns about dune erosion arose at Inchydoney, the analysis shows stable EPRs, with the 2000 shoreline more landward than the 2021 line. The most significant changes occur at the western end, where the tip of the sand spit near the estuary is retreating. These findings challenge perceptions of critical erosion while highlighting the limitations of the EPR method. The steady retreat of the vegetation line since 2012 reveals a more complex, non-linear pattern of shoreline dynamics, that could easily be missed without intermediate aerial photographs.

While these findings provide valuable data on shoreline change, they offer only a partial view. The next phase of the study will model near-shore conditions and sediment transport, and these results will be incorporated into a Coastal Vulnerability Index (CVI), assessing hazard exposure and susceptibility along the Cork coastline and linking coastal dynamics more directly to vulnerable receptors. Nevertheless, this first phase of the study provides a more nuanced and location-specific understanding of shoreline change, offering a significant improvement that enables Cork County Council to make informed decisions based on actual change data. Elementary GIS skills and minimal processing time and power are sufficient to adapt and carry out this robust and repeatable automated vegetation line detection method and produce ready-to-use and reliable change rates at a regional scale using a DSAS. The transferability of the methodology elsewhere has been proven by its ability to deal with very different coastal environments along the Cork coastline without using site-specific thresholds. The method could be readily applied at a national scale, particularly since all the datasets used provide national coverage. This method is a good illustration of Vitousek et al.'s (2023) principle, where "data-poor" archives, with spatiotemporally sparse data of disparate quality are turned into highly sought-after "data-rich" coastal science products. Another advantage of this method lies in the limited data sources needed for the analysis. The addition of ancillary data such as LiDAR and land cover, did not significantly affect the results, but did reduce processing time with less manual cleaning of the results required. While additional LiDAR and land cover datasets for each photography time period, could potentially help in rectifying minor misclassifications, the overall impact on the results is likely negligible.

## *Limitations and uncertainties*

A simple time-efficient automated method comes with limitations and uncertainties which need to be clarified and considered when using the results. Uncertainty calculations are essential when interpreting shoreline change rates, regardless of the method used to derive them. These calculations involve uncertainties related to the photography positional accuracy ranging here from 0.5 to 1 m (Table 1), and the automated measurement uncertainties, which have been estimated to be between 0.6 and 1.2 m (Table 2) with a mean 95% confidence interval of 0.98-1 m. The combination of the photography positional accuracy and the measurement uncertainties can be calculated using the square root of the sum of the two uncertainties squared (Hapke et al., 2011). This gives results ranging from ±0.6 m for the 2021 dataset to ±1.3 m for the 2000 and 2005 datasets. Finally, the resulting shoreline change rate measurement uncertainty has been estimated using a 95% confidence interval to be ±0.27 m/yr., which is once again comparable to the manual digitisation uncertainties presented by the GSI (2023). It is still valuable to draw robust conclusions from shoreline change with relatively higher error terms when calculated over longer periods where the main shoreline processes can be considered distinct from the errors (Pollard et al., 2020). The error terms presented in this study are still much lower than the ones presented in recent remote sensing studies on shoreline change with 2.37 to 7.97 m for shorelines detected with VEdge_Detector (Rogers et al. 2021) and 9.3 to 27.9 m for delineations from VedgeSat (Muir et al. 2024). The difference is largely explained by the resolution of the source images. VEdge_detector and VedgeSat are working with satellite images with coarser resolutions and therefore larger errors but over longer and denser timeseries unveiling different coastal dynamic processes. Limitations have been identified in relation to specific environments and conditions. Dune system progression can take the form of small embryo dunes which tend to be missed out by the automated method. Change rates in these environments tend to be smoothed by the method as early progression or washing away of the small dunes generally occurs. Seasonality is an important parameter to take into consideration while working with vegetation features using visible wavelengths. It is always easier to capture vegetation at its growing peak while it is at its greenest, although the timing of this may differ for different vegetation species, and indeed even between years depending on the weather. The marram grass in dune systems and agricultural grasslands in Ireland do not display the same phenology. Marram grass' green appearance is altered in July and August when it flowers, while grasslands reach their seasonal peak in these months. Late autumn and early spring photography give poorer results.

The choice of a vegetation line to serve as shoreline-proxy is not always ideal as some back beach environments might not always be vegetated, cultivated areas can be ploughed for example and these misclassifications have greater consequences if they occur on the first or last photography in the timeseries. Extra care and verification is needed in these instances. However, the vegetation line was chosen as the best proxy option for the available data and its effectiveness in detecting storm-driven changes (Pollard et al., 2020), which are a significant driver of shoreline change along the Cork coastline (Devoy, 2008). Finally, spatial resolution is a critical parameter in any remote sensing workflow. This methodology is a good illustration of the importance of recognising the fractal dimension of features of interest. An improved resolution might not always improve results, and for many sites the 0.25 m photography gives poorer results than the 1 m photography, while the 0.1 m photography gives the best outcomes due to complex vegetation edges being captured more precisely. This finding suggests that future data collection should carefully consider the optimal resolution for capturing boundary details. While higher resolutions may seem advantageous, they can introduce inaccuracies at certain levels. Therefore, a lower resolution might be acceptable for accurate boundary delineation without sacrificing detail (e.g., 1-m photography, as used in this research). Identifying the ideal frequency and timing of aerial imagery acquisition is challenging, as aerial imagery is typically collected for multiple purposes. Capturing shoreline change using a vegetation line proxy is a specific application that would benefit from annual acquisition, timed when the vegetation of interest has the greatest contrast with its background. Though the optimal timing may vary depending on the area and vegetation type, this study demonstrates that valuable insights can still be gained from aerial imagery even when acquisition conditions are not ideal.

## Conclusion

This research has demonstrated the viability of automated detection of vegetation lines on aerial orthophotography, making use of CVIs developed for very high-resolution UAV photography. The NGBDI proved to be versatile enough to distinguish the vegetation line for very different temperate coastal vegetation environments on photography with different spatial resolutions, acquired in different light and seasonal conditions. In most instances, vegetation lines extracted using the automated method are within 1 m of the manually digitised line, with a measurement uncertainty similar to that achieved by manual digitisation, even though the uncertainty of the automated method is more variable across the dataset. The uncertainty is determined to be ±0.27 m/yr. when looking at the consequent shoreline change rates, which are the much-needed end products. This automated method provides a reliable solution for local authorities and coastal managers with limited data sources, time, and remote sensing knowledge.

**Open peer review.** To view the open peer review materials for this article, please visit http://doi.org/10.1017/cft.2024.17.

**Data availability statement.** The data that support the findings of this study are available from the corresponding author, E.C., upon reasonable request, except for original data from Tailte Ireland and the OPW.

**Acknowledgements.** The authors would like to thank Eamonn Mullaly and Tomas Kavanagh from Cork County Council for facilitating access to the Tailte Eireann datasets as well as David Fahey from the Office of Public Works for facilitating early access to the latest OPW datasets.

**Author contribution.** E. C. initially devised the methodology, performed all the analysis, and led the writing of the article. F. C. contributed to method development, writing, and editing of the article, M. O.S. and J. M. contributed to the supervision of the research and editing of the article.

**Financial support.** This research was funded by Cork County Council Social Sustainability Infrastructure Programme (SSIP) and the Department of Environment Climate Change under the Cork Coastline Vulnerability Assessment project (2022–2026).

**Competing interest.** None.

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
