## [Reviewer Report]

This is a very interesting and potentially innovative research article. It covers a wide stretch of shoreline along the coast of SW Ireland. It develops a rapid automated method for extracting shorelines from aerial photographs based on vegetation rather than water lines. It has potential to allow historic shoreline analysis to be performed reliably, efficiently and to an acceptable accuracy.

Having said that I feel there are some major areas where the research could be improved. It is strange that there are two occurrences of each of the 3 figures at the end of the manuscript. I feel this paper needs more figures than just 3. I would have liked to have seen the shorelines in more detail for the different locations to get an idea of how much shoreline retreat (note: retreat NOT erosion) has taken place historically. The photographs used here go back to 2000 so cover over 20 years. Rather than just seeing the manually digitised shorelines and the automatically extracted shorelines I would like to see the shoreline from the oldest photo and the one from the newest for both methods. Then you can use DSAS to see what EPR is returned for each region using the different methods. This would also provide a context for the actual rate at which the shoreline is retreating and therefore how significant the error terms are in this context. You mention the research applies to shorelines retreating at <1 m/y in the abstract but then go on to say that the automated shoreline extraction has uncertainty of 0.27 m/y. To me this doesnt suggest a robust way of analysing shoreline change.

What about earlier time periods than 2000? Are there historic maps you can use to provide some further context on what is actually going on around the coast of SW Ireland. I am sure that would be a valuable step forward and one that you can probably do quite easily if you have the DSAS set up in ArcMap.

I would also like to see a level of engagement with the literature on shoreline proxies and how some are datum-based while others are proxy-based and where the vegetation line on the aerial photos fits into this context. Why pick this proxy-based shoreline (see the work of Pollard)? What else is available for SW Ireland. You mention LiDAR which is a great source for datum-based shorelines. How much LiDAR is there for the region? Perhaps not much, but it would be helpful to know. Also in the context, it seems a bit odd to pick out just the data for East Riding (lines 80-84) when we have so much additional data for all 6 regions around the UK coast as presented via the Channel Coast Observatory web site.

Going back to the methodology, I couldn’t really follow the selection of the CVI section where you talk about how 5 different vegetation indices perform but do not provide any detail on how you assessed the data to reach the conclusion that the NGBDI is the best one (lines 191-213).

One thing I have found when using DSAS is that the buffering and generation of the baseline is very tricky on sinuous coasts, which this clearly is. You talk about this on lines 266-269 and you mention transect casting can be difficult but did you do any manual editing and omission of transects crossing the shorelines at very oblique angles. This can generate large distances between shorelines which are not real but a product of the methodology.

So in summary I think this is an acceptable paper but it needs quite a bit of contextual work to make it more useful. How is the Cork shoreline changing over time? What data sets are available in addition to aerial photos? What other shoreline proxies could be used and why was the vegetation line chosen? Are the figures complete and why are there only 3 but 2 copies of each?

---

## [Reviewer Report]

This paper compares the performance of numerous Colour Vegetation Indices (CVI) for detecting the seaward extent of vegetation along sandy shores in County Cork, Ireland. The analysis of the performance of the CVIs is comprehensive, and the use of the Normalised green blue difference index (NGBDI) is well justified. The paper brings novelty in identifying the best index to use when identifying the vegetation line in aerial photography, which tends not to include the infrared band, which is commonly used to identify vegetation in other image types, including satellite imagery. This manuscript is well-written with few typos, but some sections are highlighted below that require more explanation. I suggest one major amendment to this paper prior to publication, as well as responses to the more detailed comments below.

Major amendment

This manuscript requires analysis and discussion on how the recorded shoreline change rates provide new insight into coastal dynamics in the region. Currently, there are no figures showing shoreline change rates across the region, and the discussion contains no information on how the identified shoreline change rates could be of benefit to coastal risk management practitioners and other stakeholders. This manuscript is using pre-existing methods and applying them to new locations, so discussion on the performance of the tool in isolation is not sufficient to be published without analysis on what new information about coastal dynamics is gained. For example, are there sections of the coastline that are more or less dynamic than previously identified? Are there dynamic regions close to vulnerable receptors? What do the observed changes tell us about the dynamics of the region?

Detailed comments:

Line 67: “...there has been growing interest from stakeholders for accurate data on coastal change…” Please find a reference specific to this claim.

Paragraph starting line 98: Other papers detecting the vegetation line using automated methods in the British Isles exist which are relevant to this introduction. The discussion would also be strengthened by considering how the performance of NGBDI compares to the methods applied in other papers.

Rogers, M.S., Bithell, M., Brooks, S.M. and Spencer, T., 2021. VEdge_Detector: automated coastal vegetation edge detection using a convolutional neural network. International Journal of Remote Sensing, 42(13), pp.4805-4835. https://doi.org/10.1080/01431161.2021.1897185

Muir, F.M., Hurst, M.D., Richardson‐Foulger, L., Rennie, A.F. and Naylor, L.A., 2024. VedgeSat: An automated, open‐source toolkit for coastal change monitoring using satellite‐derived vegetation edges. Earth Surface Processes and Landforms. https://doi.org/10.1002/esp.5835

Paragraph starting line 98: Why specifically do you choose to use the vegetation line in this study? You already highlight that each shoreline proxy has benefits and limitations, please expand further to fully justify the use of the vegetation line instead of the waterline or other proxies.

Line 104 - 105: ‘..it is an efficient indicator when looking at coastal erosion in soft sandy environments…’ Are saltmashes and mudflats present in the study area? Was the tool attempted in these environments?

Study area and Data section: How does data availability in County Cork compare to the rest of the island or Ireland? Could this study be repeated elsewhere? Did data availability contribute towards the choice of study site?

Line 164- reference to Table 1- make clear which column you are referring to.

Line 182: ‘..[this method] is not suitable for hard or artificial coasts’. Are there examples of human intervention immediately landward of the vegetation line in any of your study sites, e.g. coastal risk management features, infrastructure or settlements? It is not clear whether these locations are contained within your definition of ‘artificial coast’. Please clarify this and how the tool performed in these locations. If these sites do not contain any of these features, please highlight whether this tool is transferable to these regions. Are there examples of where the tool handles sections of vegetation line that are broken due to the presence of human interventions?

Paragraph starting line 199- How are the percentage values calculated? Please also add in values in metres to make them more meaningful.

Paragraph starting line 199- Did the other indices tend to identify the vegetation line as being more landward or seaward than the manually interpreted edge?

Paragraph starting line 199- You provide good justification on the threshold values eventually chosen for NGBDI in the following paragraph, but what range of threshold values were tested?

Paragraph starting line 223- Again, good justification of the threshold values is provided here, but please add this information to a table to make it easier for the reader. Alternatively, this information could be add to Table 2.

From a binary photograph to a vegetation line section: The inclusion of the flow diagram is useful, but please also add in an image showing the multiple processing stages to make this easier for the reader to understand and reproduce, i.e. GIS screenshots.

Results: No images of rates of shoreline change are included. Which sections of coastline are the most dynamic within each study site? Does this change over time?

Discussion: Can you add comment on the ideal frequency, location, and specification of future acquisition of aerial photography to aid this analysis?

Line 399- ‘The accuracy of the position or even the existence of a true vegetation line…” You mention here about the diverse interpretation of the vegetation line, what is the NDGDI detecting, the most seaward vegetation (excluding seaweed) or a more landward, consistent vegetation line?

Figure 3A and C- It appears that the automatically derived line flips between the most seaward vegetation and more landward vegetation in different circumstances. In general, this paper requires a tighter definition of what the ‘vegetation line’ is. It is important that the method is consistently identifying the same part of the vegetation line (see comment above)

Page 34- is this duplication of page 31?

---

## [Editor Report]

Both reviewers see a great deal of merit in the approach presented here and are encouraging about publication. But they also bring out some deficiencies that need to be addressed for the manuscript to be publishable. The reviews are thorough and Reviewer 2 identifies a series of specific points which all need to be addressed. Having said this, I don’t think any of the changes and additions asked for should be too difficult to implement. 

Some tightening (there is too much casual switching between global, England and Ireland examples throughout) and focussing (we don’t want general discussions of sea level rise rates and UAV applications but more focus on Ireland and the study area) of the Introduction is required. In particular, there needs to be a more thorough discussion of the advantages and disadvantages of different proxy-based and datum-based methods, then focusing in on the vegetation line (where there are some questions to answer on its use). Reviewer 2 asks some very detailed question under their specific comments. There is literature that should be cited (Rogers et al., Muir et al. and Pollard (which I think is in Geomorphology 354 (2020)). There is also a lot more broadscale shoreline mapping than you give credit for in a NW European context – check out the England and Wales Environment Agency and the British Geological Survey. 

In terms of Methods, the reviewers ask for more in terms of robustness and error terms, the crenulated coast problem in DSAS and walking through the analytical steps, with visual examples.

In Results, there is a need for more explanation about what the different indices are identifying and how the percentage values were arrived at. Why exactly is NGBDI best? If LIDAR made little difference here where is the analytical data to support this point? 

In different ways, both reviewers ask for more context, Reviewer 1 in terms of historical context and Reviewer 2 in terms of spatial context (i.e. location and shifts in hotspots). They both, again slightly differently, ask for more plots of actual shoreline change rates.

Finally, they would like to see more on future data collection strategies in the light of this analysis. How transferable is this methodology elsewhere when, for example, site-specific thresholds were applied? And some broader comments on how this approach informs the discussion of coastal risk in the study area (is there a massive shoreline retreat problem here?) would be valuable and more powerful than general statements about ‘global coastal decision makers’. The arguments are here, I think, but they need to be tightened and focussed. 

Overall, there is great potential here for a very valuable addition to the shoreline change literature. But some restructuring, extension of argument and attention to detail is needed to get to this point.

---

## [Reviewer Report]

This is a comprehensive revision of a really interesting and useful paper. I imagine this would be of interest to many organisations conducting shoreline change analysis. I would suggest a few minor amendments before the article is published as follows:

1. line 21-22 I would like you to justify more explicitly why you say APs are the most valuable tool for shoreline change analysis (eg: because they are historic, available, accurate, etc). You still dont make reference to historic maps.

2. line 24 develops NOT developed

3. lines 41, 56, 63, 81 in these lines you have not changed coastal to shoreline as you have done elsewhere. Was there a reason for this?

4. line 43 (see lines 21-22) - needs more justification

5. line 78 : retreat/advance NOT erosion

6. line 86-89 : you discuss the East Riding of Yorkshire but what about the whole of England where APs are available every summer going back 2 decades or more courtesy of the UK Environment Agency and available on the CCO web site. I pointed this out in my earlier review and I still find it strange that you do not mention this, especially as you conclude that this is what is needed for good shoreline change analysis.

7. line 124-133 The vegetation line may vary seasonally. You talk later about this and give information about when in the year the APs were captured (line 282 and lines 369-77). It might be better to state this here.

8. line 38 : Does NOT do

9. line 326-8 : no manual editing of transects - what error might this introduce especially on such a sinuous coast. Its OK to do this on straight coasts but I am not sure this is good practice here.

10. line 384-5 : very important point and well expressed.

11. line 438 : photographs NOT photography

---

## [Reviewer Report]

I thank the authors for revising the manuscript, which has greatly strengthened it. However, the revised version of the manuscript still does not contain enough information in the results and discussion sections pertaining to the shoreline change values observed, and what this means for coastal dynamics along these stretches of coastline. I respect the authors’ wishes to have a subsequent paper focussing on these issues, but there should still be some investigation into what the application of these methods have identified. For example, when looking at the new Figure 5, some sites are more stable than others, and some sites have differential rates of shoreline change over time or between locations. There is very little mention of this within the manuscript, or at least a discussion on some of these dynamics for a subset of the study sites. I note the authors’ rebuttal that the NGBDI index has never been used to delineate the coastal vegetation line before, but it is still a pre-existing method applied to a new application (with robust justification). Therefore, there is some methodological novelty, but my view remains that this has to be supported by some analysis of the actual shoreline change values identified using this method and their significance.

Subsequently, this paper requires further major revisions, and I have a couple of other minor comments, that are mainly typographical:

Throughout the manuscript: You sometimes use a space between a number and its metric, and sometimes do not, e.g. 10m vs 10 m. Please be consistent throughout.

Line 246- ‘recognized’. Be consistent with US vs UK spelling.

Line 326- “No manual editing or omission of transects…” If this is the case, you need to make clear that this was because there was low sinuosity in the data, if this is the case.

Paragraph starting line 371: “Seasonality is an important factor….” This paragraph is more appropriate in the discussion section.

Section 399: Validating resulting change rates- I believe the text in this section comparing EPR when using manually digitised lines vs NGBDI generated lines can be condensed. This allows space to give a few results that provide more granular detail on differences in EPR over space and time at, at least, a subset of the sites.

Paragraph starting line 464- “Prior to this work, County Cork Council…” This paragraph is a strong addition to the discussion section, as it highlights differences between observed shoreline change rates using your methods and previously identified shoreline change rates. A few other points of this nature should be included in the discussion. It may be beneficial to provide these in a separate section, but this is optional.

Figure 4- Line 785-788 the fact that the automated method flips between the most seaward vegetation and more landward vegetation is significant. Especially because the distance between these locations can be as large or larger than the EPR. Therefore, this information should be included in the body of the discussion, with further reference to the significance of this.

Figure 5- I believe this is wrongly labelled as Figure 3? Also, the lines at site B go off the top of the image, please correct this.

---

## [Editor Report]

Both reviewers recognise - as do I - that you have significantly strengthened your manuscript through the revision process. This is really commendable. We are moving rapidly in the right direction - but still not quite there.

Reviewer 1 raises some small details to attend to but also makes a couple of substantive points - i) the need to recognise the extensive (space and time) datasets of AP coverage for England and Wales that can be access through the Channel Coast Observatory and ii) the issue of lack of manual editing.

Reviewer 2 also raises some minor points that need attention. But there is also a broader criticism here which remains a point of contention. The reviewer calls for more in terms of results and the significance of those results. You and your team wish to reserve that material for a subsequent paper. I think that both these positions and fair and can be defended. But I do wonder if we could find some middle ground? 

Could you just add a little more on where you are going next? Perhaps some broad indications of where this is going. I think if you could do that, alongside the other changes asked for on the revision by both reviewers, we would then be there in terms of having a manuscript that could be well supported going forward from the handling editor. The changes will, however, need to be clear and signposted in a clear ‘response to reviewers’ statement for the handling editor.

Thank you for engaging so thoroughly with the review process and for your patience as we work our way through what is needed from here.

Professor T Spencer

Handling Editor

---

## [Reviewer Report]

The authors have addressed the comments that I have raised in previous reviews. Whilst I believe that the manuscript would be stronger if it contained results on rates of shoreline change, I am content that the authors have added further details on how this will be achieved in future research. Overall, I believe this is a very well-presented piece of work and recommend it for publication.

---

## [Editor Report]

This paper has been through a thorough review process, with two rounds of revision. I agree with the reviewer that it could be madeeven stronger but I think the requested changes have been made to a reasonable degree. My recommendation is that it is now accepted for publication.